# The Lac Fallère Area as an Example of the Interplay between Deep-Seated Gravitational Slope Deformation and Glacial Shaping (Aosta Valley, NW Italy)

Stefano Dolce ⬤, Maria Gabriella Forno *⬤, Marco Gattiglio and Franco Gianotti

Dipartimento di Scienze della Terra, Università di Torino, 10125 Torino, Italy; stefano.dolce996@edu.unito.it (S.D.); marco.gattiglio@unito.it (M.G.); franco.gianotti@unito.it (F.G.)
* Correspondence: gabriella.forno@unito.it

**Abstract:** The Lac Fallère area in the upper Clusellaz Valley (tributary of the middle Aosta Valley) is shaped in micaschist and gneiss (Mont Fort Unit, Middle Penninic) and in calcschist and marble (Aouilletta Unit, Combin Zone). Lac Fallère exhibits an elongated shape and is hosted in a WSW–ENE-trending depression, according to the slope direction. This lake also shows a semi-submerged WSW–ENE rocky ridge that longitudinally divides the lake. This evidence, in addition to the extremely fractured rocks, indicates a wide, deep-seated gravitational slope deformation (DSGSD), even if this area is not yet included within the regional landslide inventory of the Aosta Valley Region. The Lac Fallère area also shows reliefs involved in glacial erosion (*roches moutonnée*), an extensive cover of subglacial sediments, and many moraines essentially referred to as Lateglacial. The DSGSD evolution in a glacial environment produced, as observed in other areas, effects on the facies of Quaternary sediments and the formation of a lot of wide moraines. Glacial slope sectors and lateral moraines displaced by minor scarps and counterscarps, and glaciers using trenches forming several arched moraines, suggest an interplay between glacial and gravitational processes, which share part of their evolution history.

**Keywords:** Aosta Valley; glacial landforms; DSGSD; LGM; Lateglacial

## 1. Introduction

The various and complex relationship between glacial and deep-seated gravitational processes in a high-mountain environment is the subject of this study. This issue represents a poorly investigated topic due to a greater interest in the genetic mechanisms and kinematics of the deep-seated gravitational slope deformations (DSGSDs) [1–9], chronology [10–14], and current activity [15–18]; however it offers an interesting and innovative tool in the field of landscape genetics [19–22]. DSGSDs act over very long timescales [23–26] comparable, for example, to the extension of the Lateglacial period [18,27], which favours a possible interaction between glacial and gravitational processes.

Unlike most landslides, DSGSDs are characterised by slow movements of the rock mass [28–34]. Moreover, slopes affected by DSGSDs retain the original cover of glacial and gravitative deposits, even if these are partly buried by debris or extensively deformed, resedimented, and reshaped. The retreat of glaciers causes a debuttressing phenomenon that changes the stress state of the rock mass through progressive release of stress accumulated when the area was covered by ice mass [35]. The consequent propagation of joints in the rocky mass predisposes the slope to rapid gravitational readjustment or slow release of accumulated residual stresses [36–40].

In the context of a DSGSD, it is not uncommon for a glacial landform to be fragmented into slabs by one or more minor scarps (e.g., in the Pointe Leysser DSGSD) [41], while the preservation of a gravitationally displaced moraine is unusual (e.g., Rodoretto DSGSD

in the Chisone Valley) [42] or more difficult to recognise (e.g., Arp Vieille DSGSD in the Valgrisenche) [43].

More generally, both dislocation of the pre-DSGSD glacial landforms and sediments, such as glacially sculpted valley sides and moraines displaced by minor scarps or counterscarps, and glaciers that enlarged syn-glacial trenches and levelled minor scarps, suggest an interplay of glacial and gravitational processes. These processes are localised not only in the strictly DSDSD areas, but also in surrounding areas where the evidence is more nuanced.

The Lac Fallère area (Figure 1) is a sector conducive to this type of investigation, and it is characterised by a lake that shows a typical shape and a semi-submerged rocky ridge that longitudinally divides it (Figure 2). It is located in the upper section of the Clusellaz Valley, an approximately E–W trending tributary valley of the Aosta Valley. This area, despite now being totally free of glaciers, was largely shaped by glaciers during the Pleistocene, which settled a diffuse cover of glacial sediments and several *roches moutonnées*.

In addition, even if not yet included within the regional landslide inventory of the Aosta Valley Region, it shows evidence of deep-seated gravitational slope deformation (DSGSD) as scarps, counterscarps, and trenches. These evident morpho-structures involve both extremely fractured rocks of the bedrock and Quaternary succession.

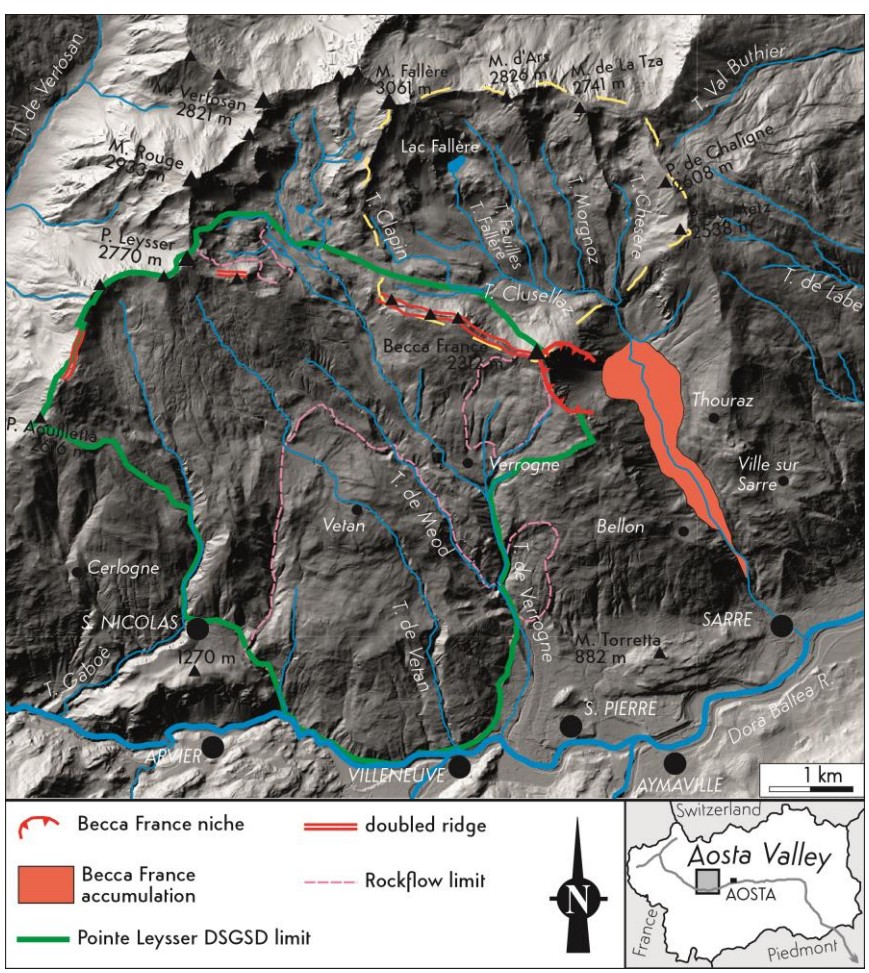

**Figure 1.** Location of the Lac Fallère area (yellow line) consisting of the upper Clusellaz Valley and its tributary valleys; The blue lines are the watercourses.

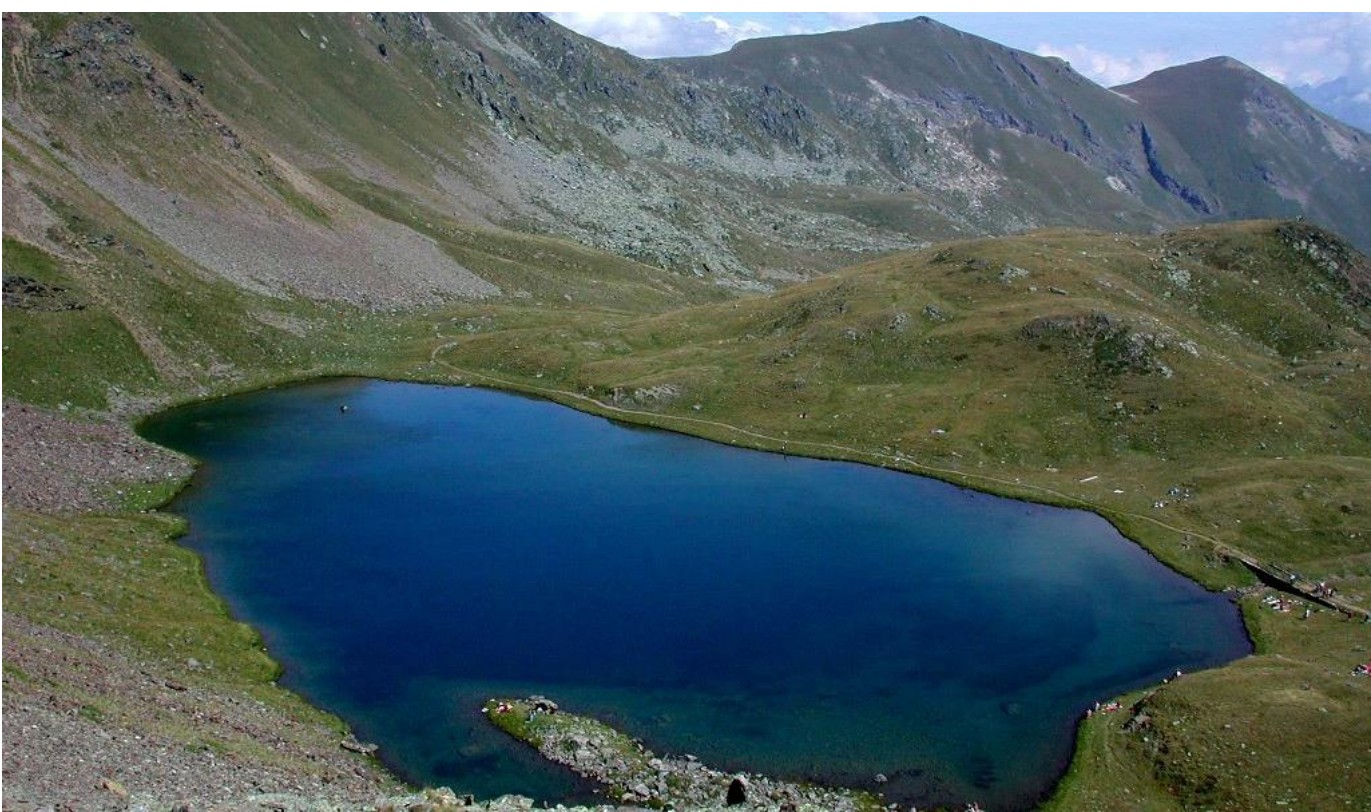

**Figure 2.** Elongated shape of Lac Fallère, along WSW–ENE trenches, and also characterised by a rocky ridge that longitudinally divides the lake.

These are very common features in the glacially shaped valleys of the Western Alps. However, the upper Clusellaz is a valley with a particularly complex geomorphology in the context of the Aosta Valley catchment due to its truncated valley head, markedly asymmetric topography in cross section, and its very wide valley floor divided into several tens-of-metres-wide parallel furrows, corresponding to a series of abandoned valleys separated by rocky ridges. Furthermore, the low watershed between the tributary valley and Dora Baltea Valley coincides with the largest doubled ridge in the region, which borders the Pointe Leysser DSGSD, an extended deep gravitational deformation affecting the left side of the Dora Baltea Valley (Figure 1). In the DSGSD area, most of the gravitational structures are evident because they show a significant post-glacial evolution (i.e., not affected by glacial modelling during the Lateglacial and Holocene periods) [44].

This case study represents an expansion of the investigations of areas even further away from the Pointe Leysser DSGSD.

## 2. Materials and Methods

A multidisciplinary approach comprising geological, structural, stratigraphic, and geomorphological methods was used, allowing for a reconstruction of the geological framework and Quaternary evolution of the study area. A geological survey concerning the bedrock and Quaternary sediments covering an area of approximately 10 km$^2$ at an altitude between 1800 and 3061 m a.s.l. was performed at a 1:5000 scale. It was integrated with photo-interpretation of the Aosta Valley 1991A (Flight RAVA 97, strip A36, photos 11–15, and strip A37-2, photos 15–18) and the use of a digital terrain model (2 m DTM 2005–2008 of the Regione Autonoma Valle d'Aosta, available on Geoportale of Valle d'Aosta, https://geoportale.regione.vda.it/download/dtm/, accessed on 1 October 2023) to better recognise and map bedrock outcrops and the distribution of Quaternary deposits, as well as major gravitational morpho-structures.

The lithostratigraphic analysis of each tectonic unit of the metamorphic bedrock was based on lithological correlation with other units outcropping in the Western Alps. The bedrock survey also identified the attitude of the regional schistosity, considered for the structural reconstruction, and structurally analysed the fracture systems. The regional schistosity, sub-parallel to the lithological layering, represents a composite foliation caused by the transposition of an older, locally still recognisable, metamorphic surface. All structural data were statistically processed through Schmidt diagrams consisting of equal-area projections plotted in the lower hemisphere using the OpenStereo programme. The Quaternary cover was distinguished using the distribution and facies of sediments and their morphological features. Due to the scarcity of outcrops, morphostratigraphy was the main method of stratigraphic subdivision.

The geological survey concerning the bedrock and Quaternary cover resulted in a new, detailed, geological map drawn using ArcGisPro software. Lithological, structural, and geomorphological data were stored in a GIS database using the WGS84 UTM zone 32N reference system, and all topographic elements were derived from the "Carta Tecnica Regionale Vettoriale" map at a 1:10,000 scale of the Regione Autonoma Valle d'Aosta (2005–2008 Edition). From the geological map, two derived, simplified maps at a scale of 1:20,000 were extracted, the first showing the Quaternary units with their geomorphological elements and the latter the gravitational morpho-structures, in which the pre-Quaternary bedrock is mapped in an undifferentiated way. The geometric relationships between Quaternary sediments/landforms and gravitational morpho-structures are represented in other, more detailed derived maps for different case studies.

## 3. Geological and Geomorphological Settings

The Clusellaz Valley is a minor (15 km$^2$) tributary valley located within the Aosta Valley, one of the wider (3262 km$^2$) mountain catchments of the Western Italian Alps, surrounded by some of the Alpine four-thousanders (Monte Bianco, Monte Rosa, Monte Cervino, and Gran Paradiso) and deeply incised.

The Lac Fallère area is included within the axial zone of the Alpine Chain (Penninic Domain) and is located along the contact between the Mont Fort Unit belonging to the Gran San Bernardo nappe system and the overlying Aouilletta Unit, referred to as the Combin Zone. These units reached alpine metamorphic peak conditions in blue schist, with variable re-equilibration in green schist [43,45].

The Mont Fort Unit consists of a pre-Variscan basement, mainly made of rusty brown, often graphitic micaschist, albitic paragneiss, Permian quartzschist, and phyllitic quartzite [46].

The Aouilletta Unit is made up of carbonate calcschist with decimetric levels of impure marble, lead-coloured phyllitic schist, and grey to white dolomitic marble with local levels of carbonate/dolomite meta-breccia [43].

The dolomitic marble, grey marble, and meta-breccia bodies were previously considered the condensed cover of the micaschist and gneiss of the Mont Fort Unit [47]. Three major, brittle fault systems developed at the regional scale intersect each other in the Lac Fallère area. The Chaligne system represents a few-kilometres-wide deformation zone characterised by sub-vertical faults trending ESE–WNW, which also locally defines the boundary between the Aouilletta and Mont Fort units [48]. The Gignod system, up to 1 km thick, is characterised by NE–SW sub-vertical faults [48,49].

Another system oriented E–W, parallel to the Aosta-Ranzola Fault, deforms the Mont Fallère southern side, as testified by faults with decametric displacement [48].

The Aosta Valley catchment consists of the 92 km long Dora Baltea Valley, into which several tributary valleys converge. The survival of 175 glaciers (covering 120 km$^2$, 3.7% of the basin) at the head of the tributary valleys is favoured by the high average altitude (2100 m) [50].

Moreover, the Aosta Valley was greatly affected by glacial modelling during the Pleistocene glaciations, which strongly modified the mountain landscape. The 130 km

long Dora Baltea Glacier repeatedly reached the Po Plain during the peaks of the 100 ka glaciations, where an imposing end-moraine system (the Ivrea Morainic Amphitheatre) was built, spanning from the MIS22-20 expansions at the end of the Lower Pleistocene [51] to the MIS2/LGM (Last Glacial Maximum) expansion at the end of the Upper Pleistocene [52–54]. According to the exposure ages of erratic boulders on the Ivrea Hills (20 ka [10]Be BP), during the Lateglacial period (19–11.7 ka BP), the glacier was completely retired into the Dora Baltea Valley, along which a series of six glacier halts (Alpine stadials) are recognised (see [54] for a review).

High average gradients (>1500 m) of the very steep valley slopes resulting from glacier modelling, coupled to post-glacial debuttressing, led to the widespread, large-scale gravitational collapse of the slopes driven by pre-existing structural weaknesses, such as faults and thrusts. These processes evolved during the Lateglacial–Holocene period at the same time as the retreat of the main glaciers forming numerous DSGSDs (279 DSGSDs covering 460 km$^2$, i.e., 14% of the whole Aosta Valley and approximately 25% of all recorded DSGSDs in the Alps chain) [2,18,55]. The post-glacial debuttressing favours along the slopes of glacial valleys both the collapse of the rock walls and the formation of major debris deposits, which partly refer to a paraglacial environment [35,56–58].

An early onset was highlighted for the Pointe Leysser DSGSD in the Mont Fallère area, following the Dora Baltea Glacier withdrawal, which triggered the collapse of the south-facing side of the Dora Baltea Valley in the Saint-Pierre, Saint-Nicolas, and Sarre municipalities [59–61].

This 23 km$^2$ DSGSD is one of the widest in the Alps chain, according to the European Alps DSGSD inventory (1033 gravitational collapses ranging from 0.03 to 108 km$^2$ with an average area of 6 km$^2$) [19,62]. The Pointe Leysser DSGSD shows an exemplar geomorphic expression due to the presence of striking 1.8 km long doubled ridges, located at 2300 m a.s.l. on the watershed between the Dora Baltea Valley and Clusellaz Valley [59]. Its prominent, most deformed 9 km$^2$ wide central sector was mapped early on as a landslide [48,63,64]. This DSGSD was also reported in the Italian Landslide Inventory (IFFI) [55]. An extension of the DSGSD in the upper Verrogne Valley (Plan di Modzon area) was later highlighted [44,65]. This outstanding phenomenon has recently been studied and proposed as a complex geosite [41]. The glacier that flowed into the Clusellaz Valley during the LGM was very extensive due to the confluence of the Verrogne-Clusellaz Glacier with several tributary glaciers (Clusellaz, Clapin, Fallère, Feuilles, Morgnoz, and Chesère glaciers) (Figure 1). The Pointe Leysser DSGSD expanded during the Lateglacial period upwards, weakening the watershed ridge between the Dora Baltea Valley and Clusellaz Valley and promoting the diversion of the Verrogne-Clusellaz Glacier, which turned south through the present Verrogne Valley (Figure 3). Gravitational deformations during the post-diversion phase largely still occurred beneath the Verrogne Glacier in the Plan di Modzon sector, where a network of glacially abraded minor scarps and trenches are masked by subglacial deposits [65]. A minor amount of deformation occurred in a glacier-free environment, e.g., at the truncated head of the Clusellaz Valley, where recognisable minor scarps and trenches are present [59].

This glacier–DSGSD interaction produced an unusually wide and gentle morphology between the Clusellaz Valley head and the middle Verrogne Valley (Plan di Modzon) that is easy to reach and favourable to the seasonal settlement of pre-historic humans at high altitudes since the Mesolithic period [44]. The local collapse of the Becca France doubled watershed in historical time (1564 AD) is considered the worst natural disaster in the Aosta Valley region, causing the propagation of a rock avalanche in the whole lower Clusellaz Valley floor (Becca France rock avalanche), where the important Thora village was settled [59].

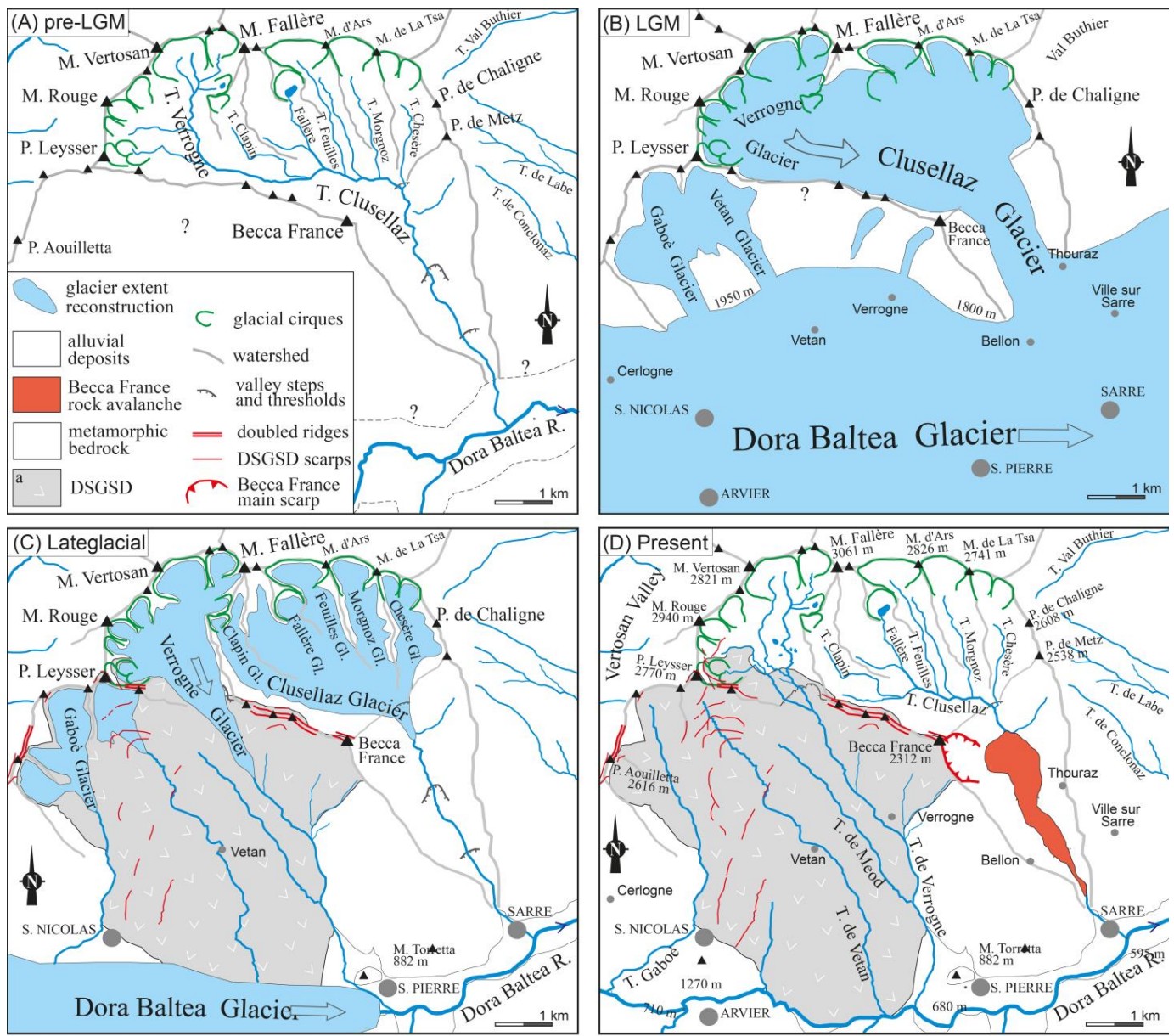

**Figure 3.** The Mont Fallère area evolution during the LGM and Lateglacial period (modified from [60]); ?: The question marks indicate the lack of knowledge on the morphology of the slope in the pre-LGM phase.

## 4. Geological Survey Results

A detailed geological survey was performed of both the bedrock and the Quaternary sediments. This survey allowed us to recognise several new pieces of information regarding the nature of the bedrock and the distribution of glacial evidence. Moreover, significant gravitational morpho-structures have been distinguished in an area not considered affected by DSGSD until now.

The geological results of this survey lie outside of the aim of this work, which is specifically focused on the relationships between glacial evidence and gravitational morpho-structures. Consequently, only simplified, derived maps from the new geological map are shown here (Figure 4).

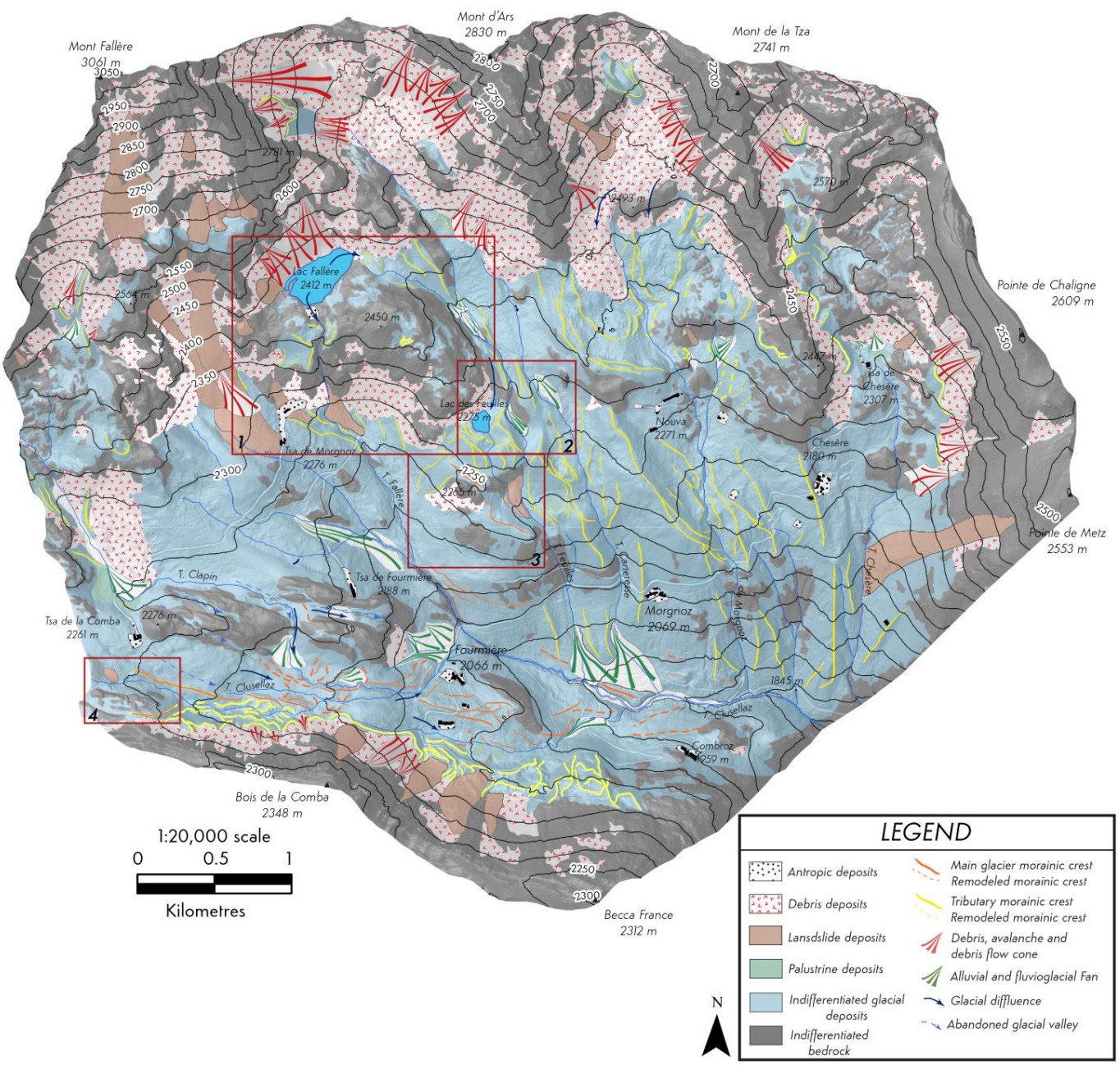

**Figure 4.** Simplified geological map of the Lac Fallère area. The red boxes refer to the case studies.

1. *Bedrock*

Two metamorphic units of the Penninic Zone are identified in the studied area: the Mont Fort Nappe, the former uppermost subunit of the Grand St-Bernard nappe system [46], and the Aouilletta Unit, referring to Combin Zone [47,66,67].

The Mont Fort Unit includes the Distulberg, Mètailler, and Greppon-Blanc formations, as defined by various authors [45,46,68–70]. The Distulberg Formation consists of micaschist, gneissic micaschist, and paragneiss. Locally, garnet micaschists were also found. Metric intercalations of quartzite, quartzite gneiss, and phylladic carbonate schists have been recognised interbedded in micaschist. The Mètailler Formation consists of various types of gneiss, mainly with chlorite, white mica, and chloritoid. Phylladic schist with graphite and ankerite in metric-thick levels is sometimes observed along the contact between the Mètailler and Distulberg formations. Both formations also host various types of metabasic rocks in metric-thick bodies with decametric extension along the regional

foliation. The Greppon-Blanc Formation consists of meta-arkose with intercalation up to metric-thick levels of metabasic rock. The top of the formation is characterised by white quartz pebbles, quartzite and bluish quartzite with albite, white mica, and ankerite.

The Aouilletta Unit is formed of various types of carbonate rocks, among which calcschist prevails. Decametric bodies of grey marble, yellowish dolomite marble, and strongly foliated beige-grey marble are present. The calcschist also contains phylladic schist and carbonate meta-breccia with centimetric to decametric dolomitic pebbles.

The bedrock is involved in four alpine deformation phases. The first two phases are characterised by pervasive schistosity transposing all previous geological surfaces, and the second phase is also responsible for developing regional foliation (S2). The third and fourth phases weakly deform the regional foliation without appreciably changing its southwards plunging.

Three major fault systems cross each other in the southern side of the Mont Fallère. Two of these brittle systems are of regional importance and deform the bedrock by a thickness of some kilometres. This particular setting can partly explain the high fracturing of the bedrock, which is made up of fractured and loosened rocks. The degree of fracturing is not homogeneous throughout the studied area, although open fractures of up to 20 cm were always observed everywhere. All fractures measured on the field were plotted in a Schmidt diagram in which it is possible to recognise some significant highly inclined ENE–WSW, NE–SW, and NW–SE surfaces and low-angle NW–SE and E–W surfaces (Figure 5).

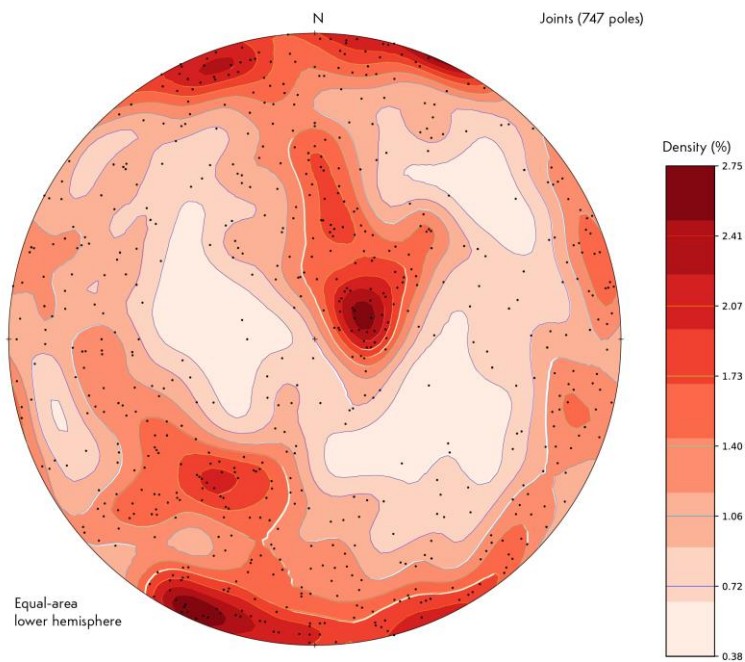

**Figure 5.** Schmidt diagram of fractures collected in the study area. The fractures are reported as plan poles (black dots). Poles are clustered according to highly inclined surfaces with ENE–WSW, NE–SW, and NW–SE trends and, at low-angle surfaces, NW–SE and E–W trends.

2. *Quaternary cover*

The detailed geological survey of the investigated area allowed us to map the glacial sediments that cover the bedrock in wide areas (Figure 4). The most common are the subglacial sediments (lodgement till, subglacial, and sub-marginal melt-out till) composed of gravel with a sandy–silty matrix that are typically massive and over-consolidated. Abundant clasts of centimetric size are subangular to angular. Their petrographic composition, formed by calcschist gneiss, micaschist, carbonate meta-breccia, and metabasite, is strictly connected to the local bedrock. The matrix is scarce and consists of unsorted, silty sand. The texture is mostly matrix-supported, but a clast-supported texture occurs locally. These

sediments cover the low and medium sectors of the slopes (with inclinations of approximately 20–25°). The subglacial sediments are connected to the different glaciers flowing on the southern side of Mont Fallère and refer to the LGM and the Lateglacial period. The overall facies of the subglacial sediments is very different from the traditional subglacial sediments that are instead characterised by the predominance of a sandy–silty matrix, an abundance of subrounded clasts with typical shapes (i.e., faceted, polished, and striated pebbles), and a petrographic composition representative of the entire basin [71].

Ice-marginal sediments are also common. They are mostly silty–sandy gravel with rare boulders up to a few decimetres in size with a clast-supported texture. Clasts display subangular to angular shapes and show a petrographic composition essentially of calcschist, gneiss, micaschist, carbonate meta-breccia, metabasite, and quartzite, representative of the local lithotypes. The scarce matrix shows an unsorted silty–sandy grain size. These sediments, which are massive or characterised by inclined bedding, appear to be normally consolidated and variously cemented.

The scarcity of large boulders, which is caused by the strong fracturing of the bedrock, represents the main difference between these sediments and the traditional ice-marginal and supraglacial ones outside DSGSD that, instead, typically show the abundance of large and giant boulders [72]. The clast petrography of the ice-marginal sediments in DSGSD areas, which is essentially local, is also different from the compositions of traditional sediments, which are usually significant for the entire glacial catchment [71]. The relatively low amounts of matrix in DSGSD areas favour the development of clast-supported textures and widespread carbonate cementation [22].

The ice-marginal sediments form several lateral and frontal moraines, which are tens of meters high and several hundreds of metres long. The location and morphology of these moraines suggest a supply both by the main Verrogne Glacier and local glaciers and a reference to the end of the LGM and, mainly, to the Lateglacial period.

The supraglacial sediments are instead very scarce, locally covering the subglacial sediments and connected to the presence of dead ice.

3.     *Gravitational morpho-structures*

Evidence of DSGSD was detected in the whole investigated area consisting of extremely fractured rocks and significant morpho-structures (Figure 6). The bedrock usually shows a joint spacing of few decimetres.

Trenches were also observed, with lengths ranging between 200 and 800 m, involving both bedrock and Quaternary successions. They are often characterised by a SW–NE trend and, subordinately, N–S, particularly closely to Lac Fallère. NNW–SSE trenches are also common to the head of the Clusellaz Valley. The shape of the lake is very peculiar, as it is characterised by a WSW–ENE semi-submerged, rocky ridge that longitudinally divides the lake and is strictly conditioned by trenches in the same direction (Figure 2).

Several minor scarps strongly characterise the morphology of the whole area, displacing the bedrock and the glacial cover. They are metres high and several hundreds of metres long and form stepped slopes of alternating rocky walls and slightly sloping sectors, where subglacial deposit occur. These sets of scarps displace glacial sediments and landforms along the original slope of the glacial valley, e.g., on the Tsa de la Comba sector.

Another typical gravitational morpho-structure is a wide, bulging relief located immediately south of Lac Fallère, where a particularly fractured bedrock appears loosened due to the presence of open fractures.

This type of structure consists of a rocky prominent and high relief separated from the main side by trenches and implies a rocky mass slid down and forward [1,73] along a basal, concave surface [74].

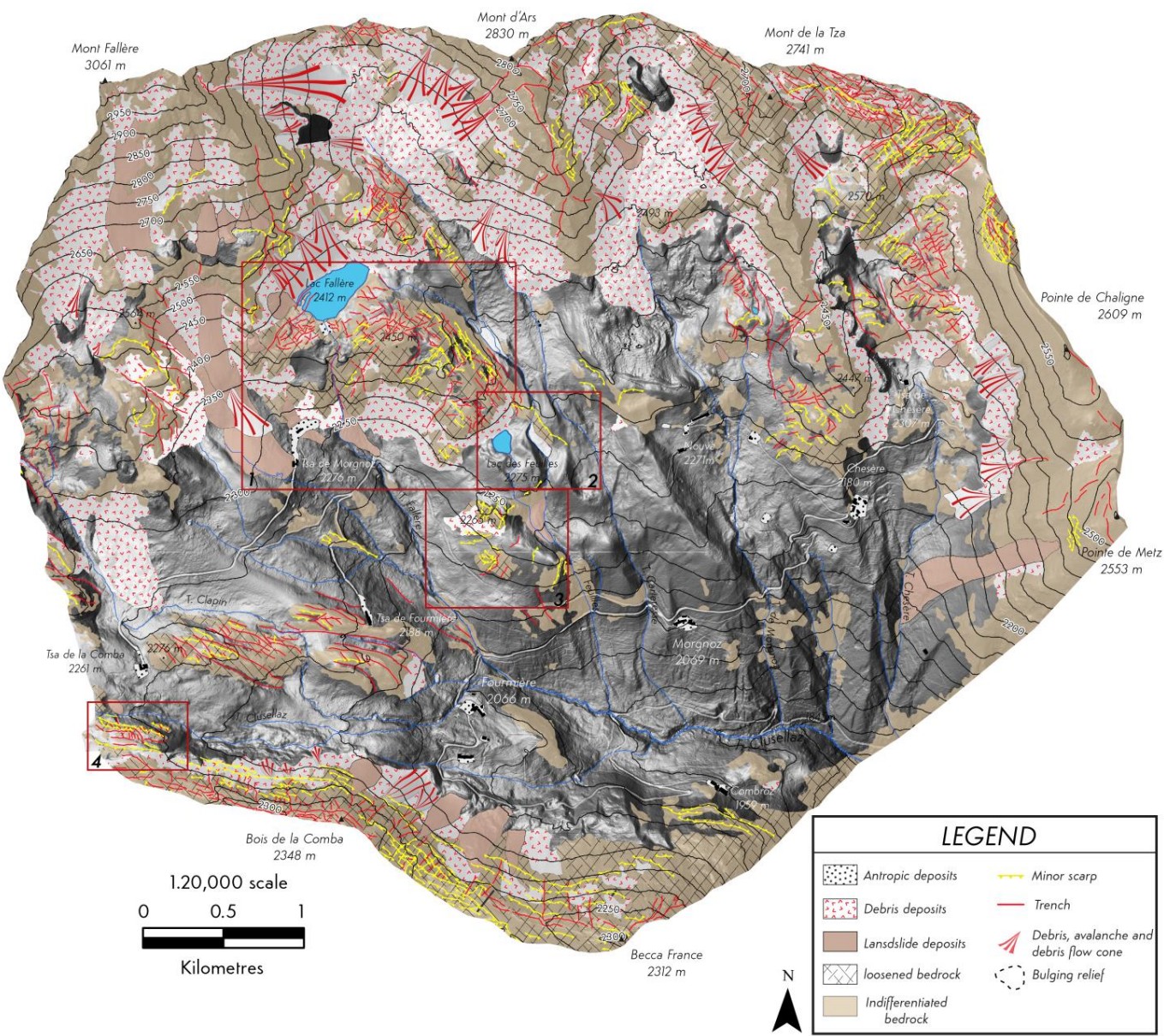

**Figure 6.** Main gravitational morpho-structures. The squares refer to the case studies.

## 5. Cases of Interplay between Glacial and Gravitational Evidence

The geological survey of Lac Fallère area evidenced several landforms connected to glacial shaping as slopes of the glacial valley covered by subglacial sediments, lateral and frontal moraines formed by ice-marginal sediments, and *roches moutonnée* due to subglacial abrasion. Gravitational morpho-structures, such as minor scarps, trenches, and counterscarps, associated with very fractured rocks are also common and involve both the bedrock and the glacial cover. The various types and ages of glacial and gravitational evidence produce different cases of interaction between glacial and gravitational landforms.

Case 1 refers to the Lac Fallère depression, which shows an elongated shape in the ENE–WSW direction, hosting a lake with a length of approximately 300 m and a width of 150 m (Figure 7). Around the lake, the micaschist of the Distulberg Formation, referred to as the Mont Fort Unit, crops out extensively. This rock consists of quartz, white mica, chlorite, albite, graphite, and sporadic garnet. They are homogeneous rocks containing small, lenticular bodies of impure quartzite and show intense brown to reddish alteration patina. Metric bodies of greenschist consisting of chlorite, albite, and actinote with gneissic structure are sporadically embedded within micaschist.

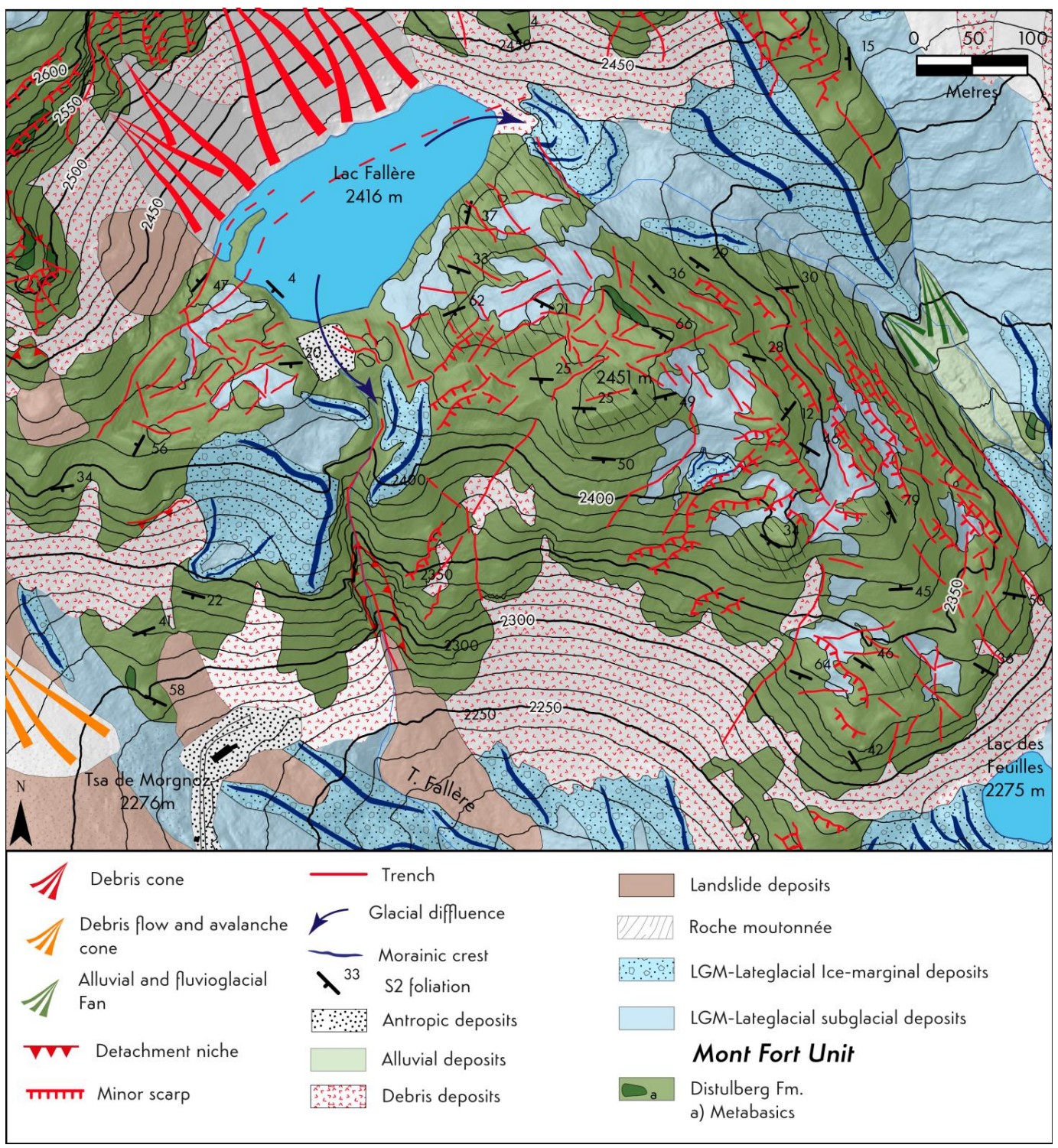

**Figure 7.** Detailed geological map of Lac Fallère (case 1) characterised by evident trenches and a bulging relief downstream of the lake. Black lines indicate the orientation of the glacial striae towards the SSE that characterise the *roche moutonnée*.

The regional foliation usually plunges on average towards the S or SW, according to the general attitude of foliation on the whole southern side of Mont Fallère. In contrast, the regional foliation results in a highly variable dip in the bulging relief (2459.2 m a.s.l.) downstream of Lac Fallère (Figure 8). The micaschist outcropping in this relief is also very fractured, with a much higher degree of fracturing than in the surrounding areas.

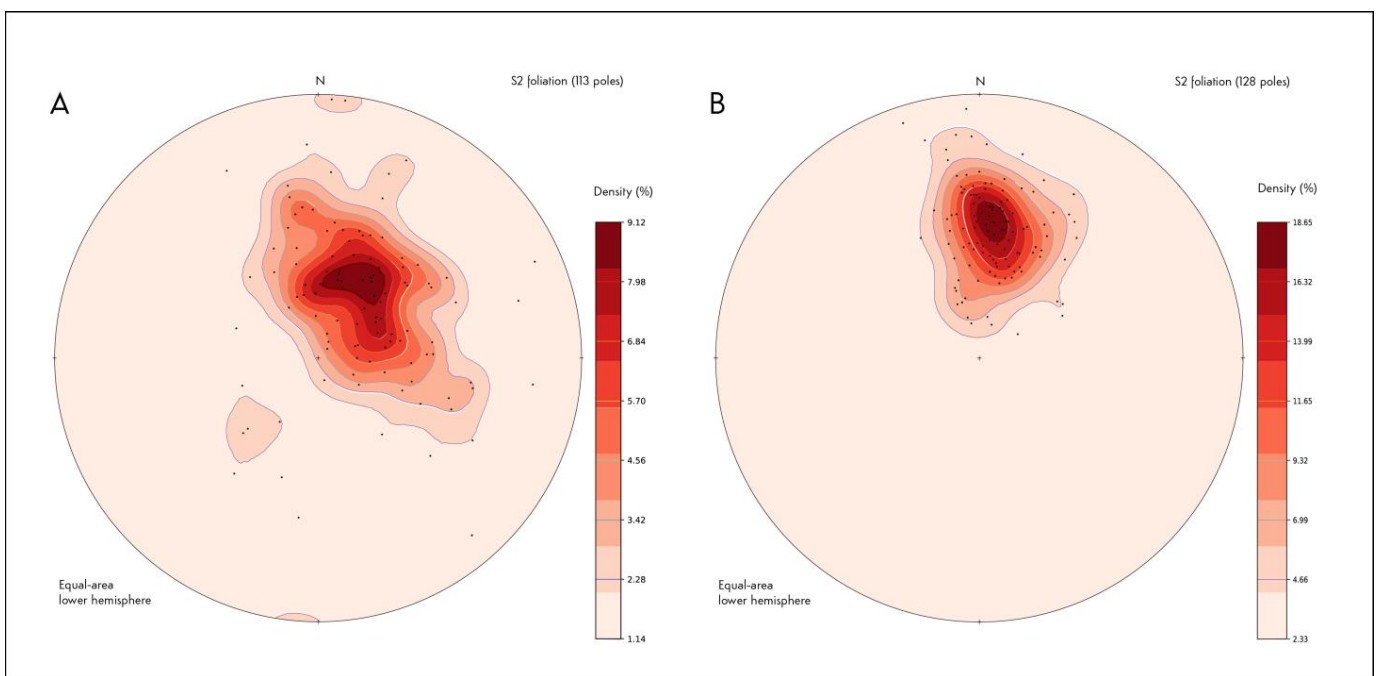

**Figure 8.** Schmidt diagrams reporting the S2 poles (black dots) for the bulging relief sector downward Lac Fallère (**A**) and in the remaining study area (**B**). The dispersion of the regional foliation S2 is very evident in the bulging relief compared with S2 outside of this.

The elongated shape of Lac Fallère occurs according to the direction of the slope and parallel to the rocky wall that defines the southern side of the Mont Fallère. A set of elongated depressions in the same direction were recognised around the lake, referred to as parallel trenches. This lake also shows a semi-submerged ridge shaped in the bedrock with the same WSW–ENE trend that partly longitudinally divides the lake (Figure 2). Lac Fallère is bordered downstream by a bulging relief with an elevation of 2459.2 m a.s.l., which is 35 m higher than the lake, locally preserving *roches moutonnée*. Several moraines are recognised along the SW and SE edges of the lake (Figure 7).

The moraines located at the edges of the lake are built by the Fallère Glacier, which formed an elongated glacial cirque strictly conditioned by parallel ENE–WSW trenches. In this case, the gravitational trenches were then possibly partly already present during the LGM and the glacier used and enlarged them, creating the depression now occupied by the lake. The early opening of the trenches before the LGM can also be assumed from the modelling of a 300 m wide SW–NE depression currently hosting the lake. Additionally, the *roches moutonnées* likely indicate a glacial shaping during the LGM. In particular, the direction of the glacial striae in the bulging relief, approximately perpendicular to the trench orientation, indicates that the Fallère Glacier flowed towards the SSE before the lake trench opening.

The location of this relief (Figure 7) downstream of the trenches associated with a highly variable dip of regional foliation and strongly fractured bedrock suggests its gravitational origin. This relief has therefore undergone more gravitational deformation than the surrounding areas and can be referred to as a bulging relief, as defined by the geological literature that described this morpho-structure in the toe sector of some DSGSDs [1,73].

The gradual evolution of the bulging relief immediately downstream of the lake, also consisting of an uplift of a rocky volume resulting from sliding on a concave surface, progressively decreased the supply of clasts to the frontal moraines of Lac des Feuilles exclusively formed by Distulberg Formation micaschist clasts. Well-preserved and relatively high frontal moraines are only present immediately downstream of the bulging relief bordered around Lac des Feuilles, which is therefore linked to the barrage by frontal moraines.

The bulging relief evolution, which probably continued during the Lateglacial period, forced the Fallère Glacier to laterally flow at its SW and NE edges, thus originating a glacial diffluence, as suggested by a significant set of moraines at these two sites. The present configuration of the lake derives from the shape of the glacier, which is in turn conditioned by the presence of trenches whose evolution likely continued in the Holocene, as testified by the semi-submerged ridge shaped in the bedrock within the lake (Figure 2).

Case 2 refers to a lateral moraine supplied by the tributary Feuilles Glacier of likely Lateglacial age. This moraine, stretched for several hundred metres in the NW–SE direction in the high basin and in the NE–SW and then again NW–SE downstream, shows an evident crest with a relatively constant slope (approximately 10°). The subglacial and ice-marginal deposits that form the moraine and its morainic crest appear interrupted by a W–E rocky scarp in the altimetric band of 2285–2300 m, which also bounds Lac des Feuilles towards the north (Figure 9). This scarp, which is shaped in the rocks from the Distulberg Formation, is not involved in the glacial erosion, an evidence that suggests a post-glacial period.

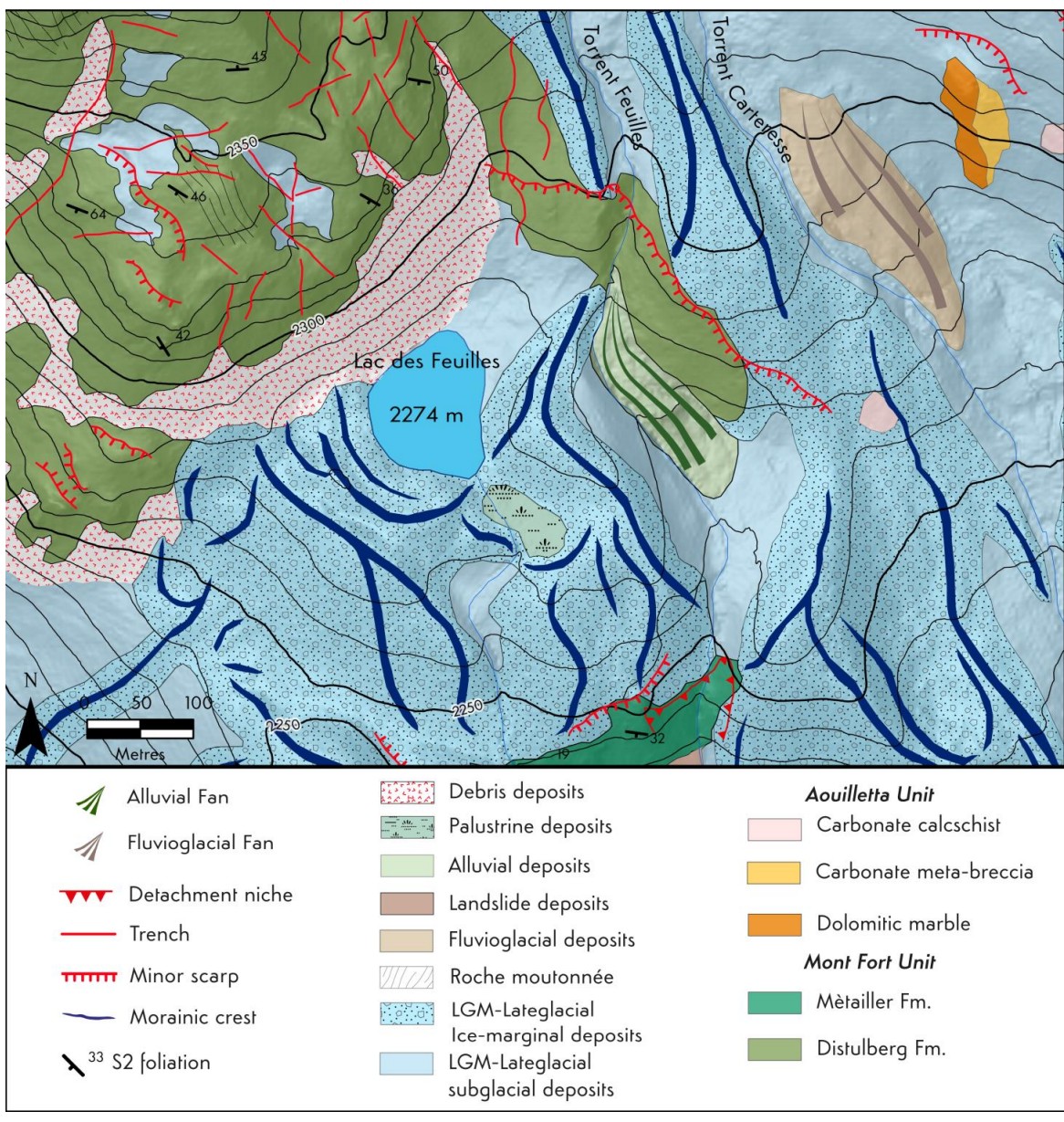

**Figure 9.** Detailed geological map of the Lac des Feuilles area (case 2) with a moraine displaced by a minor scarp immediately NE of the lake across the Feuilles T.

This lateral moraine was displaced by a rocky minor scarp and lowered of approximately 20 m in the post-glacial period. This scarp is part of the set of approximately W–E minor scarps that characterises the area and is particularly common in the sector south of Lac Fallère. The evolution of this morpho-structure is subsequent to the deposition of the more recent Lateglacial moraines (Figure 9).

Case 3 refers to the complex morphology of the rocky relief approximately 700 m north of Fourmière along the tributary Feuilles Valley (Figure 10).

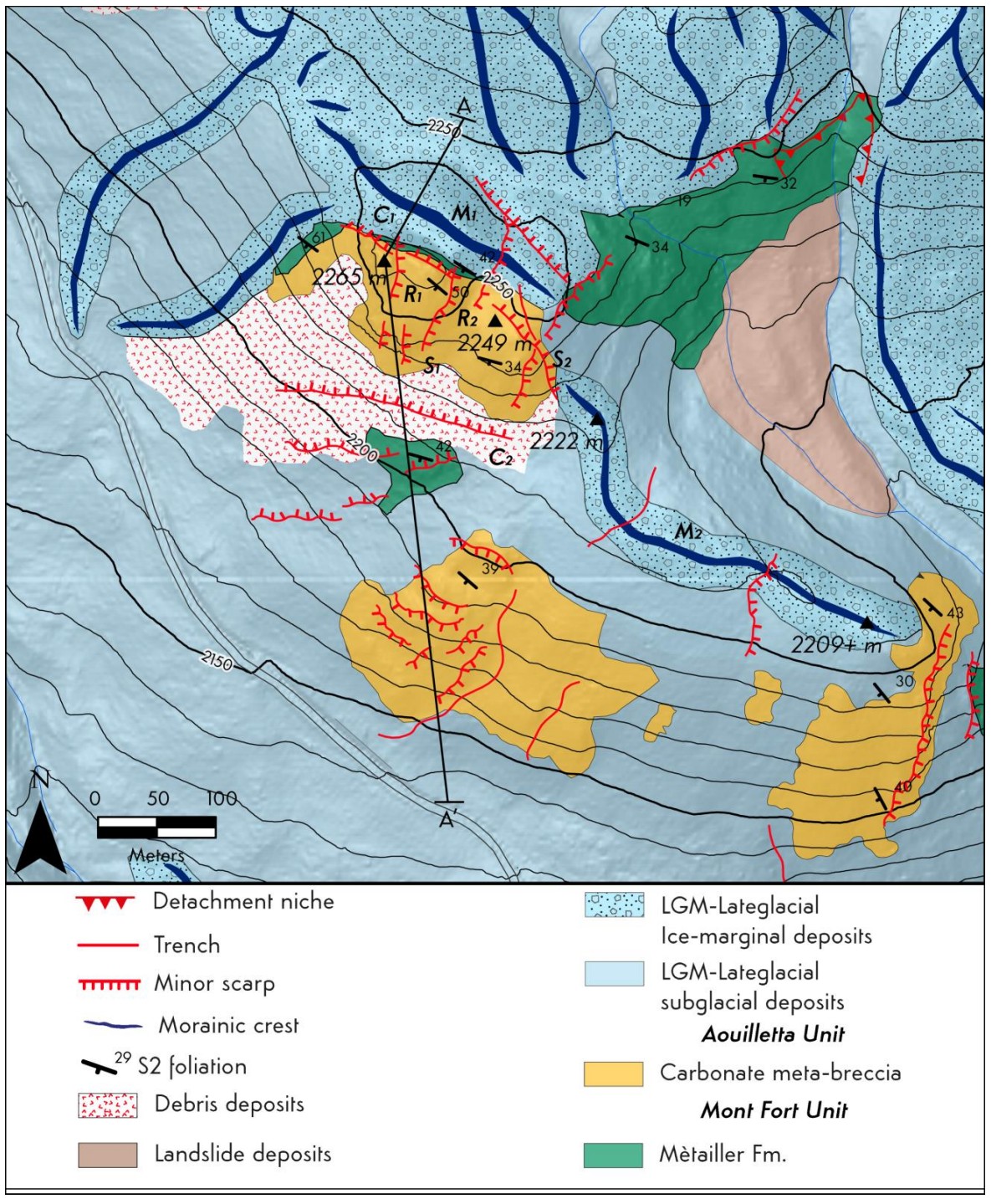

**Figure 10.** Detailed geological map of the complex rocky relief approximately 700 m north of Fourmière involved in minor scarps and counterscarps that also dislocate a moraine in $M_1$ and $M_2$. A–A$^1$ is the trace of the geological cross section.

In this area, the micaschist of the Distulberg Formation is covered by decametric, thick gneiss of the Mètailler Formation. This gneiss consists of albite, withe mica, chloritoid, rarely with sodium amphibole, and quartz showing a grey-greenish colour and schistose–gneissic structure. The contact between micaschist and gneiss plunges southward according to the regional foliation. Above the gneiss, still with a southward dipping boundary, dolomitic marble and meta-breccia with dolomitic clasts that refer to the Aouilletta Unit crop out.

This rocky relief shows an articulated shape, with a higher northern sector (2265 m a.s.l.) ($R_1$) and a lower southern sector (2249 m a.s.l.) ($R_2$), which is approximately 10 m lower. This morphology is due to two NE–SW rocky minor scarps, with heights of approximately 8 and 15 m, that dissected the relief ($S_1$ and $S_2$).

The bedrock outcropping in this relief is displaced by two gravitational morpho-structures with opposite movement to the DSGSD, referred to as counterscarps ($C_1$ and $C_2$).

The rocky relief is bounded to the north by a NW–SE moraine ($M_1$) preserved for an approximately 200 m length (Figure 10). This moraine continues downstream and appears lower, south of the relief, and with N–S and NW–SE trends ($M_2$). The dimensions, NW–SE trend, and altitude of the moraine and composition of its clasts (Mètailler gneiss with subordinate metabasite and micaschist) suggest that it is shaped by the main Verrogne-Clusellaz Glacier. The present location of this moraine north of the relief, instead of on the left side of the Fallère Valley, is likely connected to the movement along an approximately W–E counterscarp ($C_1$), which displaced this moraine towards the north and lowered it by approximately 10 m. Moreover, the trend of this moraine actually appears broken in two parts ($M_1$ and $M_2$), and the two sectors are separated by a rocky scarp. The displacement of the moraine is congruent with the displacement of the bedrock in which the gneiss/dolomitic marble boundary is lowered northward by the counterscarp $C_2$.

The two scarps ($S_1$ and $S_2$) are part of the set of approximately NE–SW minor scarps common in the investigated area, while the counterscarps ($C_1$ and $C_2$) are rarer.

The space upstream of the counterscarps (C1 and C2) represents a lowered area which was occupied by the sedimentation of the tributary glaciers, which formed several relatively thick moraines.

Case 4 is located immediately north of the Becca France relief along the right side of the high Clusellaz Valley (Figure 11). The metamorphic bedrock refers only to the rocks of the Aouilletta Unit and consists of carbonate grey calcschist with varying amounts of quartz and white mica alternating with chloritic micaceous marble showing an ochre-coloured alteration patina, in decimetre-thick layers.

Pervasive regional foliation constantly plunges to the south, with low to medium inclination.

The N-facing morphological surfaces with a constant slope of approximately 25° shaped in subglacial sediments represent sectors of the original glacial slope ($G_1$, $G_2$, and $G_3$) (Figure 11).

Their location in the main glacial valley and petrographic composition of clasts (carbonate calcschist, gneiss, micaschist, metabasite, and subordinate quartzite) indicate that these landforms are linked to the Verrogne-Clusellaz Glacier before diversion resulting in flow along the present Verrogne Valley, likely in the period between LGM and Lateglacial [41].

These sectors of the original glacial slope were subsequently displaced by two WNW–ESE rocky scarps, with heights of approximately 8 ($S_1$) and 5 m ($S_2$), suggesting an evolution of DSGSD with the formation of two minor scarps during the Lateglacial period (Figure 11).

The gravitational scarps were then locally covered by a lateral moraine formed by ice-marginal sediments ($M_1$), also referred to as the Clusellaz Glacier, as suggested by its elongation and petrographic composition (carbonate calcschist, gneiss, and micaschist, with subordinate metabasite and quartzite), because during the Lateglacial period, the Verrogne Glacier flowed along the Verrogne Valley (Figure 3).

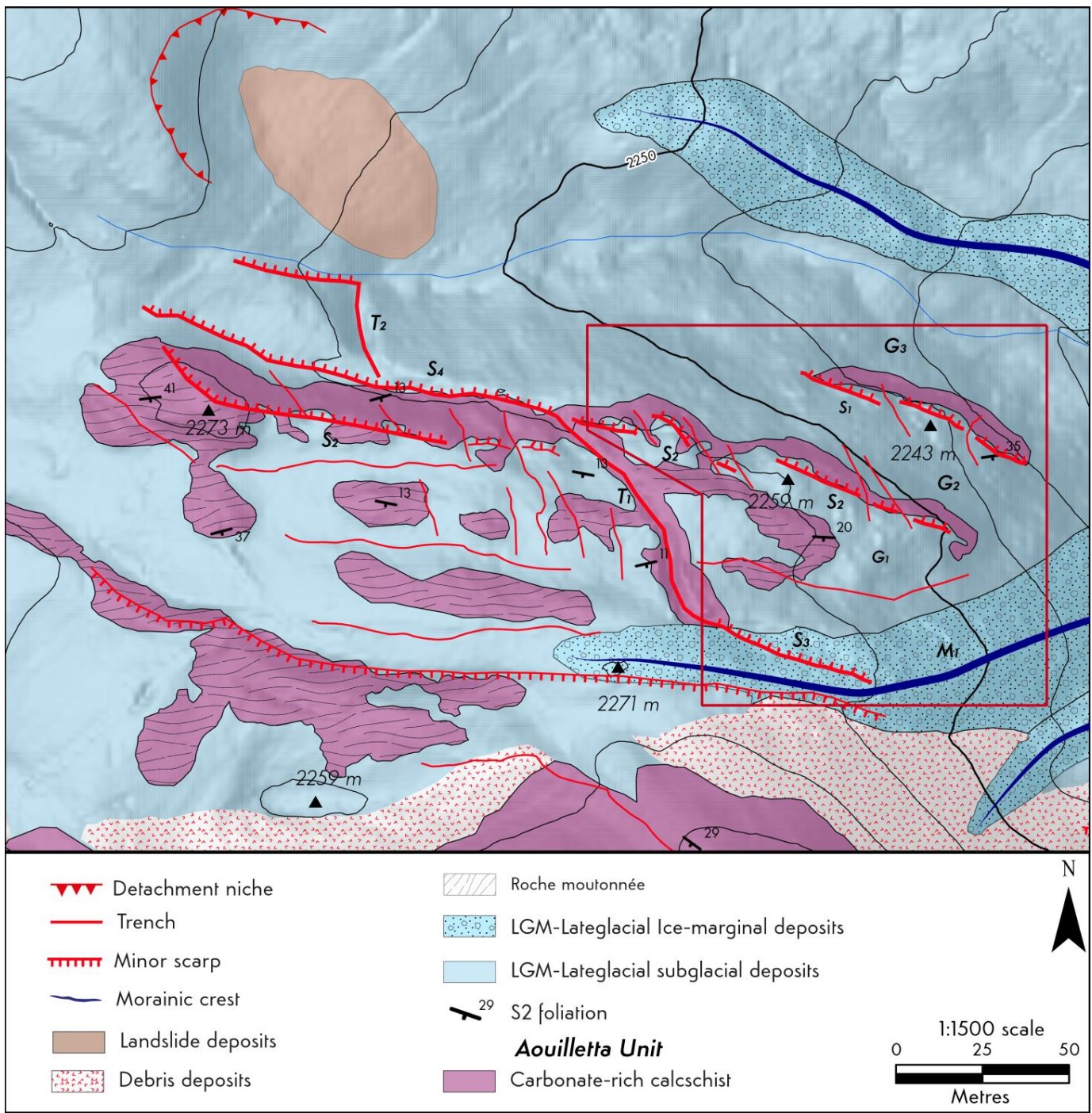

**Figure 11.** Detailed geological map of the right side of the upper Clusellaz Valley with subglacial sediments ($G_1$, $G_2$, and $G_3$) involving minor scarps ($S_1$ and $S_2$) that appear covered by a Lateglacial moraine ($M_1$). The bedrock and this moraine were later affected by a composite morpho-structure ($S_3$, $T_1$, $S_4$, and $T_2$). The square refers to a detailed picture of the relationships among morpho-structures, bedrock and glacial deposits.

This morphological arrangement is further involved in the development of a composite morpho-structure characterised by approximately E–W stretches ($S_3$ and $S_4$) consisting of minor scarps and stretches with directions between N125° and 165° consisting of trenches ($T_1$ and $T_2$). This composite landform indicates a further recent evolution of DSGSD (Figure 12).

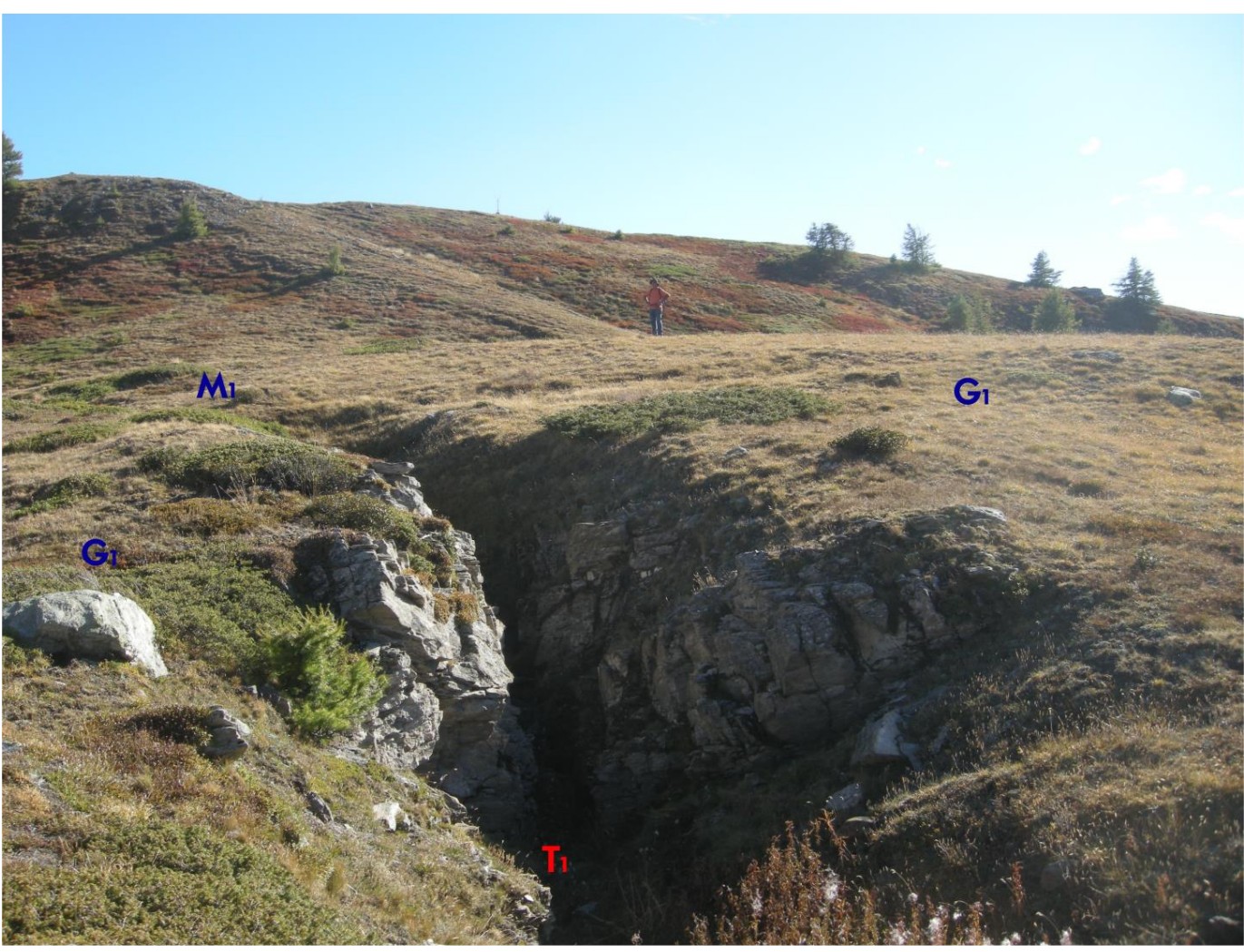

**Figure 12.** Trench $T_1$, which involves the glacial slope $G_1$ and the more recent moraine $M_1$.

## 6. Discussion

The Lac Fallère area shows evidence of glacial shaping, referred to as the Verrogne-Clusellaz Glacier and its tributary glaciers (Clapin, Fallère, Feuilles, Morgnoz, Chèsere), with different ages (LGM and the various withdrawal stages of the Lateglacial period) (Figure 4). A wide cover of subglacial sediments is preserved, essentially connected to the LGM, characterised by flat or weakly inclined morphologies. Rounded reliefs involved in glacial erosion preserving *roches moutonnées* are also recognised. The directions of glacial striae can provide information about glacier flow directions.

Several moraines linked to the different glaciers are also present. Their dimensions and distribution suggest that only the external moraines are likely connected to the LGM (Verrogne-Clusellaz Valley), while most moraines (tributary valleys) are instead referred to the Lateglacial period (Figure 4).

The Lac Fallère area also shows evidence of deep-seated gravitational slope deformation (DSGSD), evidenced here for the first time (Figure 7). Very fractured and loosened rocks are common in the entire area, which also shows particularly disjointed rocks in confined sectors. Several morpho-structures were also widely recognised, consisting of open fractures in the bedrock and minor scarps, counterscarps, and trenches shaped both in the bedrock and in the subglacial and ice-marginal sediments.

A wide, bulging relief was also recognised, which implies that rocky masses slipped down and forward (Figure 6). Displacement of the bedrock was also locally reconstructed, involving the boundary between the gneiss and the carbonate meta-breccia

(case 3) (Figure 13). This boundary is dislocated both at the cartographic scale and at the mesoscopic scale (Figure 10). Additionally, differences in the attitude of the bedrock foliation have been recognised (between the bedrock outcropping on the main southern Fallère side, where the attitude is almost constant with a dip towards the S or SW, and the bulging relief downstream of Lac Fallère, where the attitude is very variable), and these suggest the slipping of the bulging relief (Figure 8).

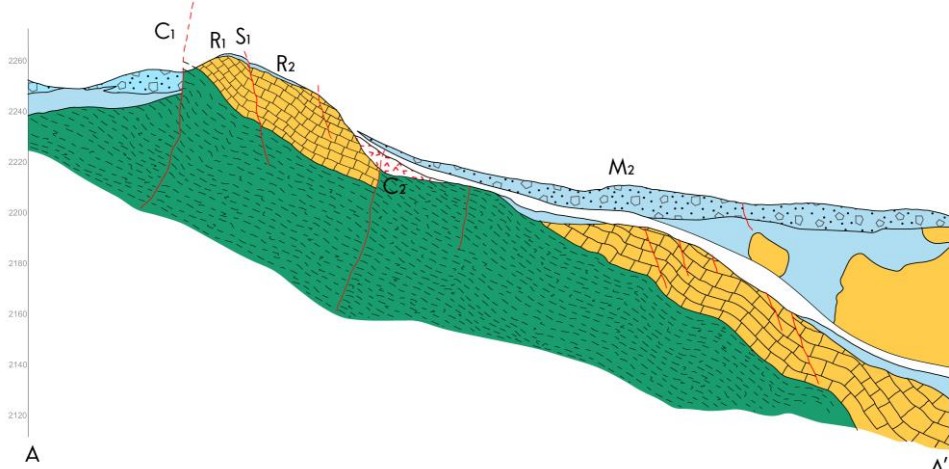

**Figure 13.** Cross section of the articulated relief north of Fourmière (case 3). The section over the white band is in the background and represents the geological view of the slope. See the map of Figure 10 for the legend and location of A–A$^1$ cross section.

Generally, the abundance of Quaternary sediments in a mountainous area can be favoured by DSGSD phenomena [44]. In detail, glacial shaping in a DSGSD environment was also produced in the Lac Fallère area, as observed in other areas [73]. This creates great effects on the facies of glacial sediments, which appear particularly rich in centimetric rocky fragments, with clast-rich subglacial deposits and ice-marginal deposits characterised by few erratics. The same DSGSD favours the construction of numerous and wide moraines, even if hosted in small basins, that are more extensive than those in areas with normally fractured bedrock outside DSGSDs [73].

The abandoned glacial valleys, separated by rocky rounded reliefs with *roches moutonnée*, favoured the construction of moraines in the whole Lac Fallère area, which are conditioned by the bedrock gravitational morphologies and not exclusively triggered by climate-driven glacier readvances.

The numerous gravitational depressions in DSGSD areas along the main trenches, which can be used for glacier flow, also favour the diffluence phenomena (case 4) (Figure 4). The separation of the Verrogne-Clusellaz Glacier into various contemporary tongues was particularly noticeable in the Tsa de la Comba sector, with the shaping of several subparallel WNW–ESE depressions reported as abandoned glacial valleys (Figure 4). The shaping of these depressions is referred to a thick Verrogne-Clusellaz Glacier essentially during the LGM and then implies a partly pre-LGM evolution of DSGSD (Figure 3). This type of depression can be interpreted as a "gravitational valley" because its shaping is partly due to the gravitational opening of trenches, as already defined in other research [73].

Exclusively during the LGM, a glacier tongue flowed across the in-uplift bulging relief, thus shaping wide areas characterised by *roches moutonnée* and constructing the Lac des Feuilles frontal moraines (case 1) (Figure 7). The distribution of glacial striae and their orientation towards the SSE on the bulging relief immediately south of the lake indicate a glacial shaping during the LGM. As the bulging relief deformation continued, the small glacial tongue of Lac des Feuilles was also depleted, stopping the flow in this direction.

Lac Fallère is contained within a WSW–ENE elongated depression and shows lateral depressions and a semi-submerged ridge shaped in the bedrock with the same trend

(Figure 2). Both depressions and the rocky ridge along the lake correspond to evidence of three side-by-side gravitational trenches (Figure 14). A thick glacier occupied the depression, due to the SW–NE trenches, which hosted an elongated cirque during the LGM (Figure 7). Consequently, the glacier enlarged the trenches likely already present before the LGM, helping to shape a 300 m wide valley. A typical cirque lake presents a sub-circular form and shows, upstream, a curved-in plant wall. The elongated shape in the WSW–ENE direction and the presence of a straight wall upstream of Lac Fallère suggest a cirque significantly affected by the evolution of trenches that therefore began their evolution before the LGM (Figure 7).

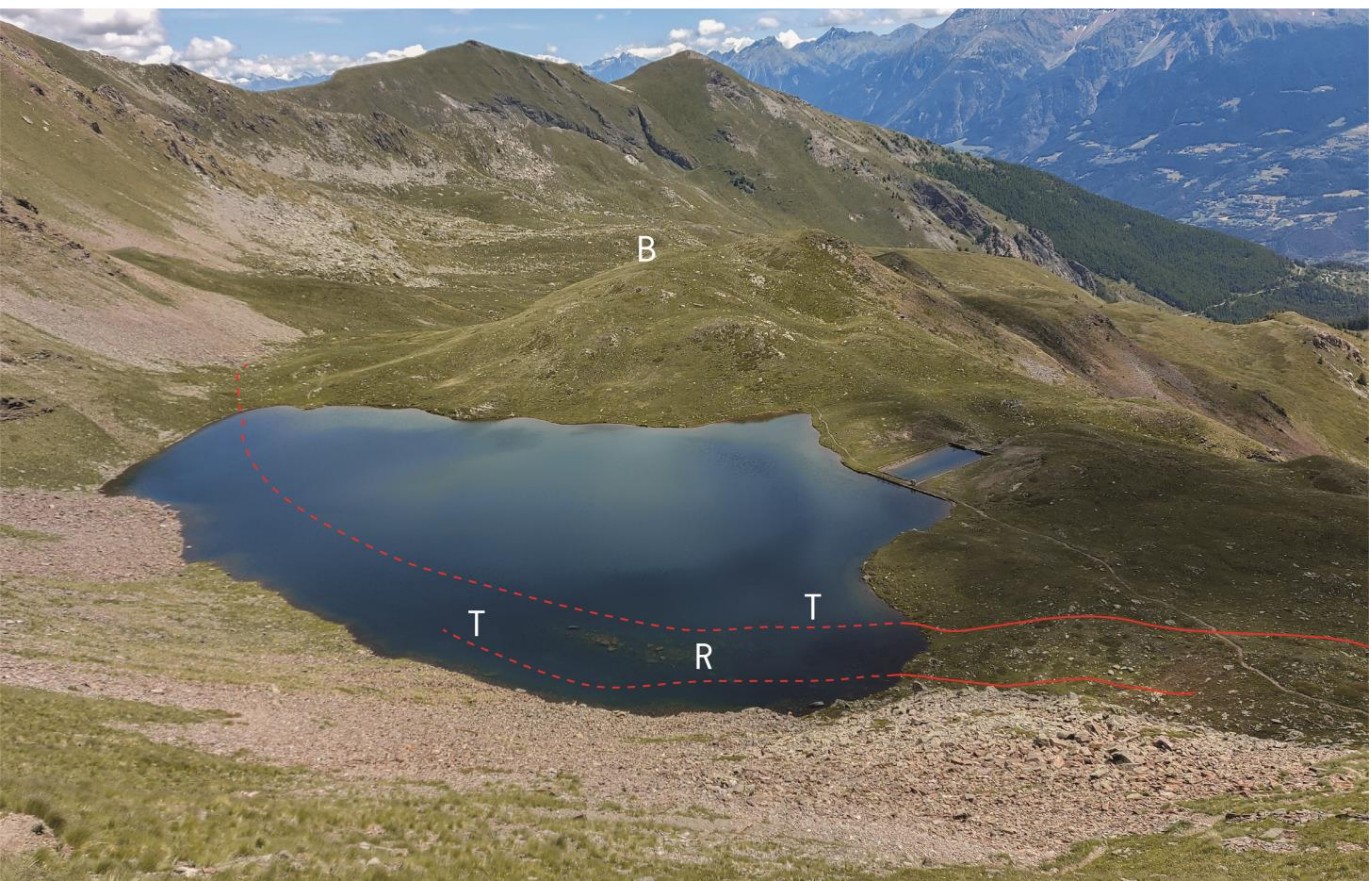

**Figure 14.** Elongated shape of Lac Fallère along evident trenches (T), bordered downstream by a rocky bulging relief (B) and characterised by a submerged ridge shaped in the bedrock (R).

The observation that locally (case 2) (Figure 9), these more recent moraines are displaced by E–W-trending minor scarps, indicates a subsequent gravitational evolution, referred to as the Lateglacial period (Figure 15). Additionally, the evolution of morphostructures, such as scarps and counterscarps, strictly influenced the deposition of glacial sediments. For example, north of Fourmière (case 3), the bedrock and the more ancient moraines were affected by dislocation from counterscarps, which defines a DSGSD evolution post-LGM (Figure 16). The presence of south dipping scarps and north dipping counterscarps defines low sectors (conceptually with graben structure) in which glacial sedimentation was favoured, with the formation of several relatively young moraines (Figure 10).

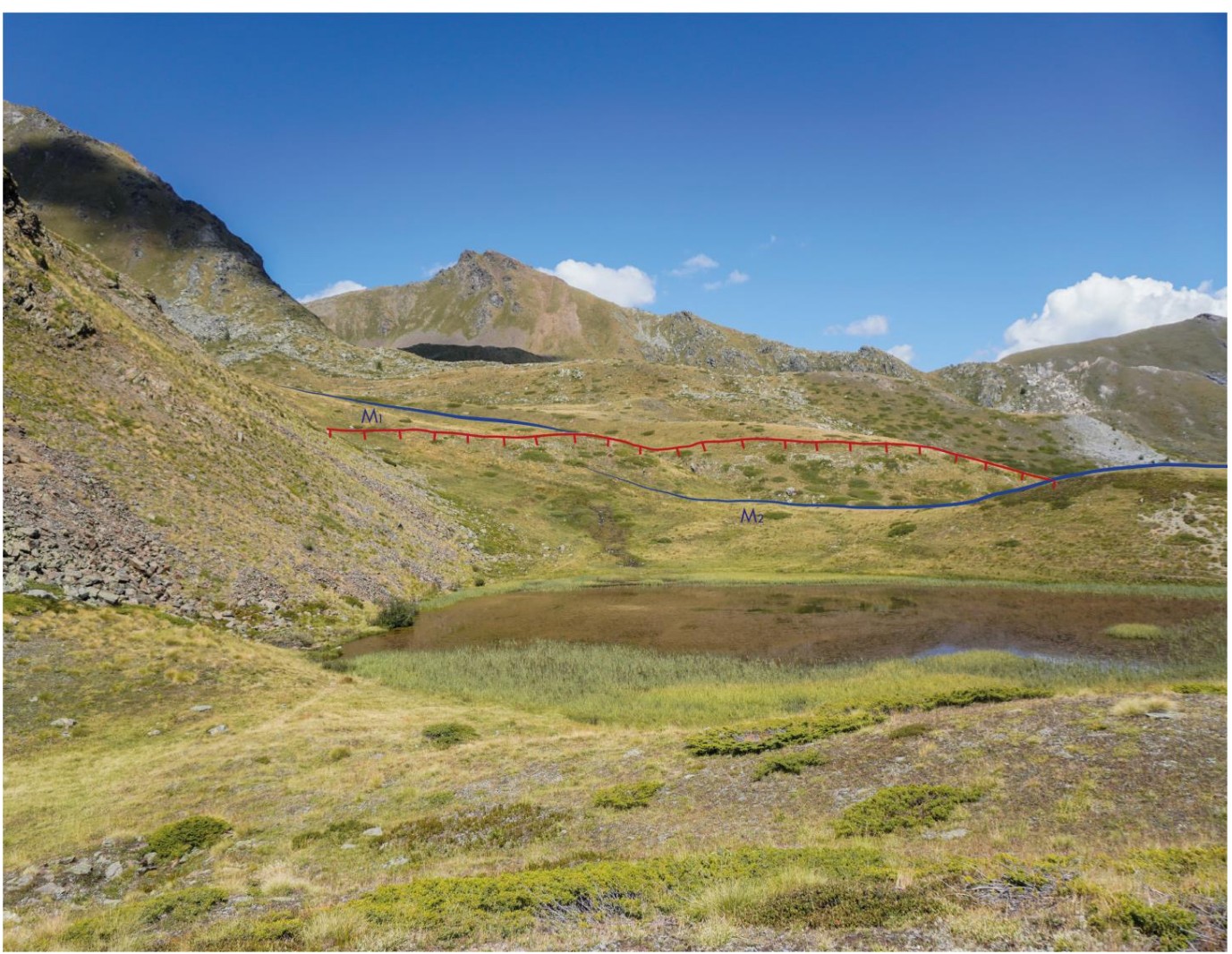

**Figure 15.** A Lateglacial lateral moraine of the Feuilles Glacier displaced in $M_1$ and $M_2$ by an E–W-trending minor scarp (red line), which exposes rocks from the Distulberg Formation.

Subsequently, another glacial diffluence occurred along Lac Fallère, where two tongues with opposite flow directions formed at the SW and SE edges of the lake (Figure 4). These two opposite tongues, which edified the more recent moraine, likely formed during the Lateglacial period along the SW–NE-evolving trenches (Figure 7).

Some stretches of the glacial side (case 4) were shaped by the Verrogne-Clusellaz Glacier essentially during the LGM because, subsequently, the diffluence of this glacier (through the Verrogne Valley) resulted in local glaciers and dead-ice no longer supplied by the main glacier (Figure 3). These glacial landforms were displaced by WSW–ENE minor scarps during the Lateglacial period (Figure 17). A subsequent Lateglacial moraine covers these minor scarps and is further displaced by a complex trench, suggesting a very recent evolution of DSGSD (Figure 12).

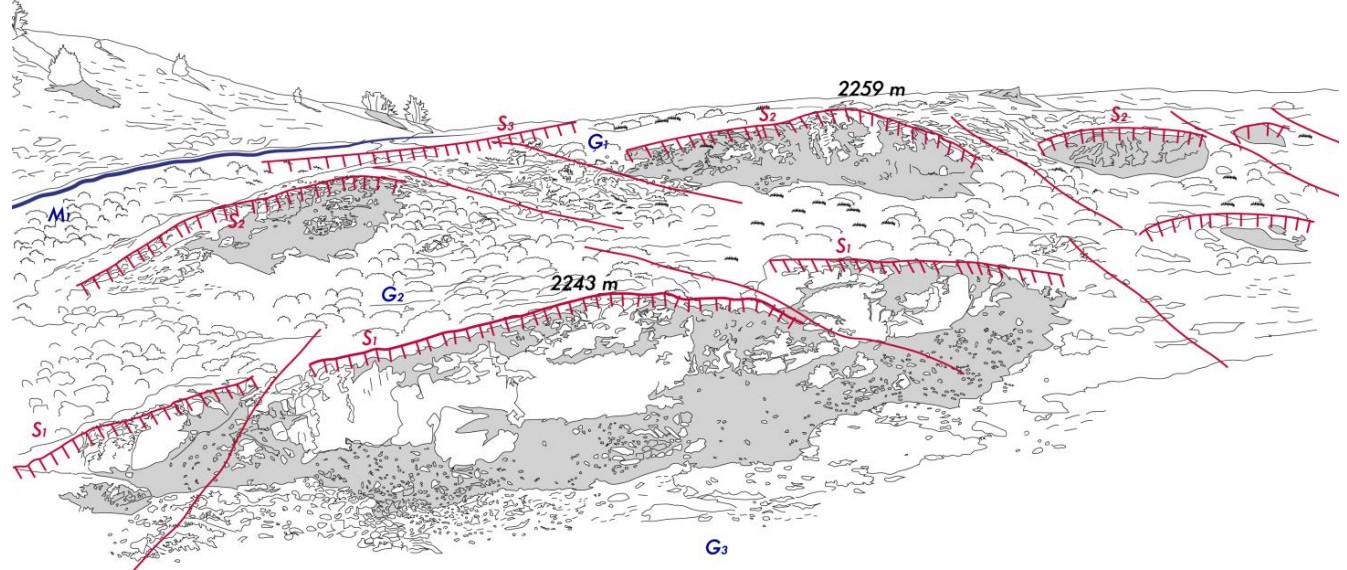

**Figure 16.** Scarps ($S_1$ and $S_2$) and counterscarps ($C_1$ and $C_2$) that dislocate both the rocky relief (with the formation of $R_1$ and $R_2$ reliefs) and the moraine ($M_1$ and $M_2$). See the map of Figure 10 for the legend.

**Figure 17.** Morphological surfaces on the right side of the Clusellaz Valley, with subglacial sediments ($G_1$, $G_2$, and $G_3$) involved in rocky, minor scarps ($S_1$ and $S_2$). The subsequent moraine (blue line) formed by ice-marginal sediments ($M_1$) covers these minor scarps and is displaced by a more recent minor scarp ($S_3$). See the map of Figure 11 for the legend.

### 7. Conclusions

The Lac Fallère area, located immediately to the NE of the large Pointe Leysser DSGSD, represents a significant example of how it is possible to preserve and identify the relationships between glacialism and gravitational structures. Such relationships are difficult or impossible to recognise in areas where the rate of rocky mass movement is high (greater than 10 mm/year) due to the intense deformation of bedrock resulting in totally disjointed rocks, with the consequent obliteration of the older DSGSD history. Very deformed rocks in the DSGSD environment can be observed in the southern sector of the Pointe Leysser DSGSD, where the disjointed rocky mass is shaped in a prominent tongue occupying the Aosta Valley floor. The rate of movement in the northern sector of the same DSGSD, as represented in the regional landslide inventory of the Aosta Valley Region, is much lower and occurs at between 5 and 10 mm/year [18]. This variable speed of rock movement is a general feature of DSGSDs that reflects their complex kinematics [17,75,76].

The polarisation of velocity in the Pointe Leysser DSGSD makes it possible to hypothesise that further north, in the area of Lac Fallère, the speed of movement of the rocky mass is lower. These conditions are therefore favourable for recognising the older history of the DSGSD in the pre- to syn-glacial time interval.

The bedrock of the Lac Fallère area is involved in three major brittle fault systems intersecting each other and developed at a regional scale. They are the ESE–WNW Chaligne System, the NE–SW Gignod System, and the E–W system parallel to the Aosta-Ranzola Fault. These regional systems are several kilometres thick and develop sub-vertical or very sloped faults, resulting in a sharp decrease in the geomechanical features of the involved rocks. The brittle regional tectonic trends are well identified in the study area through statistical analysis of the fracture data collected in the field and performed using the Schmidt diagram (Figure 5). The trends of regional brittle structures are well identifiable through this analysis, in which a SW-plunging, low-angle fracture system is also revealed, which probably refers to gravitational-deformation-related sliding surfaces.

The linear gravitational morpho-structures identified in the field also show orientation according to regional brittle trends. The concordance of orientations among regional structures, mesoscopic scale fractures, and linear gravitational morpho-structures indicates that the latter develop partly by using previous tectonic discontinuities [77,78]. Large-scale deformational morpho-structures often show features similar to brittle regional tectonic elements [62], and there is no well-defined limit between tectonic and gravitational processes [3,79].

The relationship between tectonic structures and the development of gravity morpho-structures has been modelled by some authors [74], who proposed a range of variability in the development of both morpho-structures and shape of DSGSD as a function of the type and orientation of the considered tectonic structures. Therefore, the development of gravitational morpho-structures can be considered the last step in the brittle deformation history of the orogenic chain, characterised by generalised extensional processes. DSGSD development is also strongly constrained by high relief energy.

The bedrock of the studied area, located above 2000 m a.s.l. and 1500 m higher than the main valley floor, i.e., in the morphologically highest sector of the chain, experienced a strong gravitational collapse. The ice cover present at this altitude during glaciation, which is a few tens of metres thick with a density of $0.79$ kg/dm$^3$, is probably irrelevant in influencing the forces driving the gravitational collapse of a bedrock with an average density of $2.7$ kg/dm$^3$. As a consequence, gravitational structures could develop independently with respect to the presence of the glacial mass. Gravitational structures that developed before the LGM were often completely obliterated by glaciers, probably constraining their flow directions as well. Gravitational structures developed before the LGM were reused by glaciers, as in case 1, and are still recognised if subsequent gravitational deformation was not particularly developed. Post-LGM gravitational structures are easy to recognise as they interrupt glacial deposits and landforms, as in case 2.

In conclusion, this research reports gravitational morpho-structures that cut glacial evidence and glacial sediments and landforms that seal morpho-structures, suggesting long activity of the DSGSD from the pre-LGM to the post-Lateglacial period.

**Author Contributions:** Conceptualization, M.G.F. and M.G.; methodology, F.G.; software, S.D.; validation, M.G.F., M.G. and F.G.; formal analysis, S.D.; investigation, S.D.; resources, S.D.; data curation, F.G.; writing—original draft preparation, M.G.F., M.G. and F.G.; writing—review and editing, M.G.F., M.G. and F.G.; visualization, S.D.; supervision, M.G.F. and M.G.; project administration, M.G.F. and M.G.; funding acquisition, M.G.F. and M.G. All authors have read and agreed to the published version of the manuscript.

**Funding:** The project was supported by the University of Torino ("Ricerca Locale ex 60% 2022 and 2023", grants to A. Festa), the Italian Ministry of University and Research ('Cofin-PRIN 2020 "POEM project—POligEnetic Mélanges: anatomy, significance and societal impact", grants no. 2020542ET7_003 to A. Festa).

**Data Availability Statement:** The datasets presented in this article are not readily available because the data are part of an ongoing study. Also, the original contributions presented in the study are included in the article.

**Conflicts of Interest:** The authors declare no conflict of interest.

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
