# Peer review of "The Lac Fallère Area as an Example of the Interplay between Deep-Seated Gravitational Slope Deformation and Glacial Shaping (Aosta Valley, NW Italy)"

_2624-795X, doi:10.3390/geohazards5010003_

Round 1

Reviewer 1 Report

Comments and Suggestions for Authors

please see my full Comments - if you would like any fuller discussion on how your interesting perceptions might be developed, within the paraglacial and rebound-stress frameworks, I would be very happy to do so.

if you do want to develop this work, I would strongly encourage you to involve a glacial geomorphologist to map and interpret your area, and a geo-engineer to model the ice loading/unloading---bedrock erosion rebound---slope-stress interactions.

Comments on the Quality of English Language

the English is pretty good but I could have offered numerous minor editorial improvements - again, happy to assist here with any revised submission.

Author Response

The authors of the paper thank the reviewer for the careful re-reading of the paper. We are field geologists, specialist in survey of metamorphic bedrock and Quaternary deposits and landforms. Consequently, we place great emphasis on field work, using observations of aerial and satellite photos to supplement the data. Therefore, we believe that the first section of the work concerning the geological setting of the area and results of the geological survey are essential for interpreting the geological-structural evolution of the area. We also think that this section is preliminary for assessing the relationships between glacial shaping and gravitational evolution. The four case studies, although small in size, represent significant examples of the relationships between glacial modeling and gravitational evolution.

Choosing to investigate the topic “relationships between glacial modeling and gravitational evolution” was suggested by the topic “landslides” of the special volume of GeoHazards.

This paper represents a first contribution on glacial shaping of the area based on terrain data and it is our purpose to expand the research to include geophysical studies.

We thank you for reporting some contributions on the topic of "paraglacial" which we cite.

The text and references related to general work on DGPVs have been expanded.

The case studies refer to surface dislocations but in the context of morpho-structures involving strong rocky thicknesses (e.g., several hundred meters observed in the Becca France detachment niche) and therefore these features can be considered in the context as DGPVs.

Figure 3 is only intended to depict the evolution of glacialism in a significant surrounding of the studied area while detailed glacial evidence is depicted in a specific detailed map (Figure 4).

The formation of the graben is related to extension (as added to the conclusion) which does not imply any uplift along counterscarp.

Case 1

We interpret the sector located SE of the Fallère Lake as bulging relief and not as a simple landslide because it is characterized by loose rock with variable bedding and a set of minor scarp and trenches. It is our opinion that the bulging was caused by movement of the rock mass on a basal listric slip surface.

Case 2

The scarp that displaces the moraine is not involved by the glacial erosion suggesting a developing subsequent to the deposition of the moraine.

Case 3

On the field, it is evident that the surface dipping upstream and named as C1 in Figure 10 shows no evidence of glacial shaping and can be read as a counterscarp that caused the lowering of the M1 moraine related to Verrogne Glacier.

The association, along the slope, of scarps and counterscarps with opposite dip allows the location of the M1 moraine within a graben-like structure.

Case 4

On the field, the wide fracture (Figure 12) continues in the scarps S3 and S4 and therefore cannot be considered a karst feature.

As the authors have recognized in other areas of the Western Alps, trenches can have the characteristics of wide open fractures or shallow depressions when filled by glacial deposits.

Reviewer 2 Report

Comments and Suggestions for Authors

I have reviewed the manuscript “The Lac Fallère area as an example of the interplay between 2 deep-seated gravitational slope deformation and glacial shap-3 ing (Aosta Valley, NW Italy)”.  I think the manuscript is complete and well displayed. The authors present impressive cases of DSGSDand relationships with glacial processes. I congratulate the authors for the great detail and emphasis in the geological and structural analysis and for the great quality of the images, geological cross sections and maps. The results are comprehensive, interesting and easy to read.

 -Recommend adding contourlines in the geoplogical map (figure 4). This would help read the landforms.

 -Check that all figures are referenced in the text.

 Beyond these small recommendations I consider the paper acceptable for publication

 I can only congratulate the authors for the manuscript.

 I wish you good work!

Author Response

The authors of the paper thank the reviewer for the careful re-reading of the paper. We are grateful for the very positive comment of the text and for appreciating the quality of the images, geological maps and cross sections.

We followed the suggestion to adding contourlines to the geological map of figure 4, also adding contourlines to figure 6.

We also check that all the figures are mentioned in the text. We thank also for this suggestion.  

Reviewer 3 Report

Comments and Suggestions for Authors

The article is well written and I suggest it be published after minor revisions.

Some suggestions are as follows:

1. It is recommended that the order of Figure 2 and Figure 1 be reversed.

2. Line 82: What are the sources of DTM data? 

3. Figure 8: The words on the Figure 8 cannot be read clearly.

4. I suggest separating the conclusion from the discussion.

Author Response

The authors of the paper thank the reviewer for the careful re-reading of the paper. We are grateful for the very positive comment of the text.

  1. We followed the suggestion to change the order of figure 1 and 2;
  2. We specified the sources of DTM data;
  3. We increased the font of the words in figure 8;
  4. We separed discussion from conclusion.

Round 2

Reviewer 1 Report

Comments and Suggestions for Authors

1. no additional photographic or satellite imagery has been provided as requested, to demonstrate convincingly that the alleged DSGSD terrain features involve significant displacements of bedrock.

2. taking case 2 as a key example, neither the satellite imagery nor the Aosta DTM (now helpfully linked in the text and closely scrutinised by this reviewer) suggest any dislocation of the moraine as alleged. The authors insist in their response that the lack of glacial moulding of the step-scarp is evidence that it has displaced the moraine.  However the 'freshness' of this little scarp could be for several reasons (glacifluvial erosion, in-situ rockwall crumbling).  The only photo (Figure 16) is taken side-on from a distance and is unable to show any displacement.  A continuous moraine severed as alleged would have shorn ends, whereas this moraine suite is uniformly smooth on the imagery and DTM.

3. the DTM in fact shows areas of 'blistered' landscape very similar to the study area all across this broad cirque floor complex and in adjacent parts, suggesting there is nothing unusual here to merit a paper devoted to a few quite trivial study sites.  A paper which examined all such 'blistered' areas in a much broader study area might produce more useful data and a basis for comparisons and investigations.

4. the authors assert that the features they map must be deep-seated because Becca France is (the well-known nearby rockslide cavity).  Such speculation is wholly unscientific.  The 'null hypothesis' must be that all the surface irregularities are either due to local differential erosion or (if fractures and minor displacements are verified) that they are shallow (eg. sub 10-20 m into bedrock).  Becca France  is in an entirely different topographic context, on an exposed trough flank, whereas the study sites are in lesser relief within a cirque basin.

4. the authors state that 'it is our purpose to expand the research to include geophysical studies'.  Until such studies are complete, this paper should remain unpublished, as a preliminary working hypothesis.

5. the authors now emphasise that the phenomena they have visually mapped are conventional gravitational spreading and rock-mass sliding (even with a listric sliding base !) - whereas DSGSD generally lacks such bases because it has not begun to slide as an organised mass.  DSGSD is just that - a 'deformation'.  This suggests that the authors do not have a good understanding of RSF (Rock Slope Failure) and landslide geomorphology in general.  (RSF is now the standard term in international use embracing all modes of rockslide, rock slope deformation etc. - no mention of it in the text)

6.  for example, key reference 35/56 (Ballantyne) is cited with respect to 'debuttressing' as a generator of slope stresses.  As the authors themselves inadvertently state, 'debuttressing' applies primarily to steep trough walls (as Becca France) - not to shallow cirque floors !

7. the authors did not pick up on the original review suggestion (item 2) that 'postglacial rebound' could account for rock mass displacements such as they posit here.  Additional references have been offered to them as a further encouragement to pursue such a line of analysis - which would be much more original and could justify a closely-focussed pilot study here.

the following additional references should be considered, in relation to rebound phenomena

Persaud, M. & Pfiffner, O.A. 2004. Active deformation in the eastern Swiss Alps: Post-glacial faults, seismicity and surface uplift. Tectonophysics, 385, 59-84.

Ustaszewski, M.E., Hampel, A. & Pfiffner, O. 2008. Composite faults in the Swiss Alps formed by the interplay of tectonics, gravitation and postglacial rebound: an integrated field and modelling study. Swiss Journal of Geosciences, 101, 223-235.

Jarman D, Harrison S (2019): Rock slope failure in the British mountains. Geomorphology 340, 202-233.(section 4)

Comments on the Quality of English Language

use DSGSD consistently (not DPPV)

Author Response

(The authors gave the same response as above.)
